# Individual stochasticity in the life history strategies of animals and plants

**Pablo José Varas Enríquez**[1,2,3]⊛*, **Silke Van Daalen**[1,4]⊛*, **Hal Caswell**[1]‡

**1** Institute for Biodiversity and Ecosystem Dynamics, University of Amsterdam, Amsterdam, The Netherlands, **2** Department of Human Behavior, Ecology, and Culture, Max Planck Institute for Evolutionary Anthropology, Leipzig, Germany, **3** BirthRites Independent Max Planck Research Group, Max Planck Institute for Evolutionary Anthropology, Leipzig, Germany, **4** Biology Department, Woods Hole Oceanographic Institution, Woods Hole, MA, United States of America

⊛ These authors contributed equally to this work.
‡ HC is the senior author on this work.
\* pablo_varas@eva.mpg.de (PJVE); silkevandaalen@hotmail.com (SVD)

**Data Availability Statement:** All files are available at the OFS repository that can be accessed through the DOI 10.17605/OSF.IO/5DHKM.

**Funding:** This research was supported by NWO-ALW Project ALWOP.2015.100 awarded to SvD

## Abstract

The life histories of organisms are expressed as rates of development, reproduction, and survival. However, individuals may experience differential outcomes for the same set of rates. Such individual stochasticity generates variance around familiar mean measures of life history traits, such as life expectancy and the reproductive number $R_0$. By writing life cycles as Markov chains, we calculate variance and other indices of variability for longevity, lifetime reproductive output (LRO), age at offspring production, and age at maturity for 83 animal and 332 plant populations from the COMADRE and COMPADRE matrix databases. We find that the magnitude within and variability between populations in variance indices in LRO, especially, are surprisingly high. We furthermore use principal components analysis to assess how the inclusion of variance indices of different demographic outcomes affects life history constraints. We find that these indices, to a similar or greater degree than the mean, explain the variation in life history strategies among plants and animals.

## 1 Introduction

The life history of any organism can be described as movement among a set of stages, with individuals moving at rates that depend on the phenotype and the environment. The stages may be defined in terms of age, size, developmental stage, or other criteria. Any demographic model for the life history includes at least three processes: development (transitions among the living stages), mortality (transition out of the set of living stages), and fertility (production of new individuals). Each of these processes is stochastic. Mortality or survival happen with a probability. Reproduction, and the number and kind of offspring produced, are random. Transitions during the developmental process are equally governed by probabilities.

Stochastic transitions lead to variation among individuals in longevity, reproduction, and development, even if they experience identical rates and risks at every stage. This variation is said to be due to individual stochasticity [1]. It is critically important to distinguish the

and by the European Research Council under the Seventh Framework Program through ERC Advanced Grant 322989, the European Union's Horizon 2020 research and innovation program through ERC Advanced Grant 788195 awarded to HC. Also, PVE was funded by the CONICYT Programa de Formación de Capital Humano Avanzado/MAGISTER BECAS CHILE/2017 - 73180230. The funders had no role in study design, data collection and analysis, decision to publish, or preparation of the manuscript.

**Competing interests:** The authors have declared that no competing interests exist.

variation resulting from individual stochasticity from variation due to heterogeneity among individuals. A demographic model describes the rates applying to all individuals in a given stage. Heterogeneity describes variation due to differences among individuals in the vital rates they experience in a given stage. Calculations of variance from a demographic model explicitly assume that all individuals experience the same rates at any life cycle stage, and variances resulting from individual stochasticity provide a critical standard of comparison in the search for evidence of heterogeneity (e.g., [2–5]).

Population biology and human demography have long focused on expected values of the outcomes of demographic models, and these expected values are usually considered as life history traits. Life expectancy, for example, is the expectation of longevity; the net reproductive rate $R_0$ is the expected value of lifetime reproduction; and the generation time is the mean age at reproduction. But expected values give only part of the story, and it is now possible to calculate the variation, due to individual stochasticity in the life cycle, of many life history outcomes.

The variability implied by a life cycle is calculated by embedding the population projection matrix into a finite state, discrete time, absorbing Markov chain. Such a chain includes transient states corresponding to living stages of the life cycle, and one or more absorbing states corresponding to death or other ways of leaving the living states. Longevity is the time required to reach an absorbing state; its mean, variance, and other moments are calculated from the Markov chain. Lifetime reproductive output is calculated by associating a 'reward' with every transition in the Markov chain. The reward represents stage-specific reproductive output; the Markov chain provides the mean variance, skewness, and all the moments of lifetime reproductive output, starting from any living state. For examples of Markov chain methods applied to individual stochasticity (or what would now be called individual stochasticity), see [1, 3, 4, 6–13].

Quantifying the variance in demographic outcomes is important because any analysis of a random variable is incomplete until it includes some indication of variation. In the case of life histories, variance due to heritable differences among individuals in fitness-related traits is potential raw material for natural selection. Variance due to individual stochasticity is not, so it is important to know how much variance is implied by the life cycle, as a null model for evaluating potential for selection [12]. In applied contexts, planning and management require some quantification of risk (as actuaries and investors are well aware) and risk can be measured by variance, skewness and other measures of variability.

Because variation in demographic outcomes is implied by, and can be calculated from, a demographic model, it should be treated as a life history trait [14] and incorporated into comparative life history studies, just as the mean values are. This study provides the first detailed cross-taxon analysis of the variation in demographic outcomes, treated as life history traits, and how those measures of variation relate to other life history traits, across species.

We will consider individual stochasticity in a range of demographic outcomes (lifetime reproductive output, longevity, age at maturity, measures of iteroparity, and others). They will be calculated using Markov chains and Markov chains with rewards, and applied to population projection matrices obtained from the COMADRE Animal Matrix Database and the COMPADRE Plant Matrix Database [15, 16].

Our first goal is to quantify the amount of variation in demographic outcomes within populations for a large collection of plant and animal species. We will explore the distribution among populations of those measures of variation. Then we will examine how the measures of variation fit into the patterns of association among life histories, by extracting Principal Components that arrange species along life history axes. These axes can be interpreted in terms of combinations of life history traits, as has been done before for *r*- and *K*-selection (a binary

contrast) [17], the fast-slow continuum (a one-dimensional arrangement) [18], and more recent two-dimensional arrangements [19, 20]. Our results reveal patterns that are not apparent in analyses restricted to mean properties.

The paper is organized as follows. In Section 1 we describe the sources of our demographic data, and in Section 2 we describe the calculation of the statistics of demographic outcomes. In Section 3 we report the patterns of variability for lifetime reproductive output and longevity, and in Section 4 we present the life history patterns emerging from a multidimensional approach; we end with a discussion in Section 5. Details of the calculations, and supplementary figures, are presented in S1 Appendix.

## 2 Data

A comparative analysis of demographic outcomes requires demographic information for a diverse set of species. To obtain this information, we use the COMADRE and COMPADRE matrix databases. These databases are open-data online repositories of high-quality, parameterized matrix population models for animal (COMADRE v2.0.1) and plant (COMPADRE v4.0.1) species (obtained from www.compadre-db.org [15, 16]). As the databases are of public access, they do not contain information that could identify or harm the populations that are part of our sample.

From these databases we selected those matrix population models that represented natural (i.e. unmanipulated) environments. We limited our analysis to matrices that were defined as an arithmetic average of matrices defined for a single time point (as provided by the database), with an annual projection interval, with non-zero fertility values, and with at least five stages. We eliminated two-sex models because they create life cycle pathways that do not appear in the more common single-sex models. We also eliminated matrices including clonal reproduction because they would require a separate analysis of lifetime reproductive output to separate clonal and sexual reproduction.

Calculations of lifetime demographic outcomes make frequent use of the fundamental matrix $\mathbf{N} = (\mathbf{I} - \mathbf{U})^{-1}$ (see a further definition of $\mathbf{N}$ in section 3). We excluded matrices for which the condition number of $(\mathbf{I} - \mathbf{U}) > 1000$; these matrices are near-singular (the species is near-immortal) and the fundamental matrix is unreliable.

Examining the matrices, we found that some life histories are modelled in a way that creates enormous values of the statistics of LRO (e.g., variances on the order of $10^{11}$). This often results from an unfortunate choice by the investigators of how to combine, within one life cycle, the production of huge numbers of seeds or larvae and a miniscule fraction of these probabilities that survive to enter the life cycle. To prevent these models from artifactually dominating all of our analytical results we applied a method of [21] to screen for and remove outliers. The method is designed specifically for skewed data and identifies robust upper and lower bounds defined in terms of the quartiles of the data. Let $Q_1$, $Q_2$, and $Q_3$ be the 0.25, 0.5, and 0.75 quantiles of the data. The upper limit defining outliers is $Q_3 + 2c(Q_3 - Q_2)$. After some experimentation on our part, we chose $c = 5$ as a value that identified a group of genuine outliers. We excluded matrices for which the mean, variance, or kurtosis of LRO exceeded this value.

These criteria resulted in sets of 83 animal matrices and 332 plant matrices from the COMADRE and COMPADRE databases respectively, representing 47 animal and 141 plant species. The animal matrix models represent mostly age-structured populations, with model dimensions ranging from 5–69 (mean = 12.2). The plant matrix models are mostly size-structured or based on ontogeny, with model dimension between 5–26 (mean = 7). The biases in geographic, phylogenetic, and life history coverage of the matrices in these databases have been described elsewhere [15, 16].

## 3 Calculation of individual stochasticity in life history traits

Life history traits in common usage include those related to survival and longevity (e.g., life expectancy), to the amount of reproduction (e.g., $R_0$, total fertility rate), and to the timing of reproduction and its distribution over the lifespan (e.g., age at maturity, generation time, modes of parity). Our approach differs because we will compute not only the mean of these traits, but also the higher moments and measures of variance, skewness, and kurtosis.

These results are obtained from the population matrices. The database supplies, for each population, a transition matrix $\mathbf{U}$ and a fertility matrix $\mathbf{F}$. The $(i, j)$ entry of $\mathbf{U}$ is the probability of surviving and moving from stage $j$ to stage $i$. The $(i, j)$ entry of $\mathbf{F}$ is the mean number of stage $i$ offspring produced by a stage $j$ individual in a unit of time. The matrix $\mathbf{U}$ forms the basis of a finite-state, discrete-time, Markov chain. The life cycle contains $\tau$ transient (i.e., living) stages and $\alpha$ absorbing (i.e., dead or removed) stages. The transition probability matrix for the chain is

$$\mathbf{P} = \left( \begin{array}{c|c} \mathbf{U}_{\tau \times \tau} & \mathbf{0}_{\tau \times \alpha} \\ \hline \mathbf{M}_{\alpha \times \tau} & \mathbf{I}_{\alpha \times \alpha} \end{array} \right) \tag{1}$$

where the $(i, j)$ entry of $\mathbf{M}$ is the transition probability from the $j$th living stage to the $i$th absorbing stage. The total number of states is $s = \alpha + \tau$.

From the Markov chain matrix $\mathbf{P}$ and the fertility matrix $\mathbf{F}$ we calculate life history traits as follows. Details of the calculation are given in S1 Appendix.

### 3.1 Longevity

The longevity of an individual is the time required to leave the set of transient states [1, 7]. A key piece in all the calculations is the fundamental matrix $\mathbf{N}$, given by

$$\mathbf{N} = (\mathbf{I} - \mathbf{U})^{-1}. \tag{2}$$

The $(i, j)$ entry of $\mathbf{N}$ is the mean time spent in stage $i$, starting from stage $j$, before eventual absorption (death). We calculated the first four moments of longevity, starting from the first stage in the life cycle, using the expressions in Eq (21)–(24) in [22]. From these moments we calculated the mean, standard deviation, coefficient of variation (CV), skewness, and kurtosis of longevity.

### 3.2 Lifetime reproductive output (LRO)

We calculated the statistics of LRO using Markov chains with rewards (MCWR, [4, 8, 13]). This analysis treats reproduction as a reward collected by an individual at each step in its life, up until death. In the absence of data on the distribution of individual stage-specific reproduction, we used a Poisson distribution to create the reproductive reward matrices. For species that produce multiple types of offspring (i.e., in which $\mathbf{F}$ contains positive entries in more than one row), we summed all types at each age and treated the sum as the mean stage-specific reproductive output. We computed the first four moments of LRO according to Theorem 1 in [4]. From these moments we calculated the mean, variance, standardized variance, skewness, and kurtosis of LRO. The standardized variance in LRO; i.e., the ratio of the variance to the

square of the mean, was calculated to account for the opportunity for selection (OFS), following [23]. Here, he shows that the standardized variance in fitness is the rate at which selection would increase fitness if all the variance were genetic.

### 3.3 Distribution of reproduction: Generation time and modes of parity

Reproduction happens at different ages during the life of an individual. The distribution, over the lifetime, of reproduction provides two useful demographic outcomes: generation time and a measure of parity.

The cohort generation time is the mean age at which offspring are produced by an individual. The formula for age-classified models is classical (e.g. [24]), but we need a version that applies also to stage-classified models. The necessary result is found in Appendix A in [1]; see also Chap. 5 in [25]. When multiple types of offspring exist, we summed all types of offspring at every stage before calculating the generation time.

The distinction between semelparity and iteroparity (reproducing only once vs. being capable of reproducing repeatedly) focuses on the extent to which reproduction is spread over the life cycle. The degree of parity is difficult to analyze demographically because of the rarity of models that explicitly include parity (see [26] for an example of an age-parity model). Based on recent calls for a continuous approach towards parity [27], we calculated the modes of parity by extending the calculation of generation time to compute the variance, and thence the coefficient of variation of the ages at which offspring are produced. Strict semelparity at a fixed age (as in the periodical cicadas *Magicicada sp.*) would have a CV of 0. If reproduction is spread over many ages, the CV will be greater than 0 and measure how widely spread the ages at reproduction are. The derivation of this measure of parity is given in S1 Appendix.

### 3.4 Age at maturity

We calculated the mean, standard deviation, and coefficient of variation of the age at maturity using the methods of Section 5.3.3 in [7]. This involves creating an absorbing set consisting of all reproductive stages, and calculating the moments of the time required to reach that absorbing set.

### 3.5 Measures of uncertainty

The measurement of the variability of any trait requires some thought. In addition to the variance and related statistics, we calculated the skewness and kurtosis of longevity and of lifetime reproductive output, as measurements of the uncertainty of individual fates. Skewness is a measure of asymmetry of the distribution, and high values of skewness imply that the variable is even less certain than might be expected from the variance. Kurtosis measures the 'heaviness' of the tails of the distribution, and thus gives an indication of the likelihood of extreme values. The variance, skewness, and kurtosis were calculated from the moments using the formulas from [4, 8].

### 4 Individual stochasticity generates variability: How much?

In this section we explore some of the patterns generated by individual stochasticity in the statistics of lifetime reproductive output and longevity. We focus on these patterns because they are important components of fitness. Longevity integrates the chances of survival over all the pathways in the life cycle. LRO integrates both survival and fertility. Our preliminary analysis showed that both of these quantities are highly variable, both within and between the populations in our subset.

**Table 1. Quantiles of lifetime reproductive output (LRO) and longevity for the subsets of animals and plants.**

| LRO | $Q_{2.5}$ | $Q_{25}$ | Median | $Q_{75}$ | $Q_{97.5}$ |
|---|---|---|---|---|---|
| Animals | | | | | |
| Mean | 0.11 | 0.60 | 1.14 | 2.01 | 6.60 |
| Variance | 0.26 | 2.10 | 6.41 | 22.00 | 201.08 |
| OFS | 1.09 | 3.04 | 4.92 | 9.94 | 90.37 |
| Skewness | 1.00 | 2.35 | 3.37 | 4.45 | 13.19 |
| Kurtosis | 0.08 | 6.61 | 15.27 | 25.37 | 216.85 |
| Plants | | | | | |
| Mean | 0.02 | 0.30 | 1.09 | 3.20 | 18.72 |
| Variance | 0.07 | 6.18 | 67.57 | 1416.0 | 38073 |
| OFS | 2.12 | 13.06 | 75.96 | 584.8 | 6486.9 |
| Skewness | 2.26 | 5.37 | 13.71 | 35.49 | 133.28 |
| Kurtosis | 6.92 | 38.86 | 244.4 | 1674.8 | 24087 |
| **Longevity** | | | | | |
| Animals | | | | | |
| Mean | 1.16 | 2.99 | 4.57 | 6.93 | 31.08 |
| St. Dev. | 0.71 | 3.35 | 5.38 | 8.57 | 28.03 |
| CV | 0.56 | 0.88 | 1.07 | 1.25 | 1.89 |
| Skewness | 0.34 | 1.99 | 2.31 | 3.08 | 9.01 |
| Kurtosis | -1.25 | 4.32 | 7.30 | 11.10 | 79.91 |
| Plants | | | | | |
| Mean | 1.03 | 1.99 | 3.26 | 6.81 | 41.98 |
| St. Dev. | 0.50 | 2.25 | 5.15 | 11.76 | 105.53 |
| CV | 0.37 | 0.87 | 1.26 | 2.12 | 4.72 |
| Skewness | 0.66 | 2.34 | 4.31 | 9.16 | 53.71 |
| Kurtosis | 2.15 | 8.01 | 26.59 | 133.86 | 3852.7 |

Individual stochasticity creates a tremendous amount of variability in lifetime reproductive output in both animals and plants. Table 1 shows the median (preferable to the mean as a measure of central tendency for skewed data) and quantiles (2.5%, 25%, 75%, 97.5%) of each of the statistics of LRO and longevity. The 25% and 75% quantiles define the interquartile range, capturing the central 50% of the observations. The 2.5% and 97.5% quantiles capture 95% of the observations, and can be treated as the "range" of variation without the instability of minimum and maximum values. The scaled measures of variation (CV, standardized variance, skewness, kurtosis) are particularly useful indices for comparative purposes, because they are dimensionless.

## 4.1 Lifetime reproductive output

In animals, the median standardized variance (opportunity for selection) in LRO is 4.92, and is typically (i.e., 50% of the cases) between 3.04 and 9.94. In plants, the median is 75.96, with typical values ranging from 13.06 to 584.8. LRO is positively skewed; in animals the median skewness is 3.37, with an interquartile range from 2.35 to 4.45. In plants, the median is 13.71 and the typical range is from 5.37 to 35.49. For comparison, the skewness of the exponential distribution is 2, which shows that individual stochasticity typically generates distributions of LRO that are more skewed than the exponential.

Kurtosis is seldom included in measures of variability, but it is important in these life cycles. Kurtosis is scaled relative to that of the normal distribution. The median kurtosis among

animal populations is 15.27 (6.61—25.37), and among plant populations the median is 244.4 (38.86—1674.8). The tails of the distribution of LRO are thus far heavier than those of the normal distribution.

Plants life cycles create more individual stochasticity than do animal life cycles, and the ranges in variance, skewness, and kurtosis in LRO are greater among plant species than among animal species.

## 4.2 Longevity

In studies of longevity, coefficient of variation is preferred over standardized variance as a dimensionless measure of variation. The median CV of longevity in animals is 1.07 with typical values ranging from 0.88 to 1.25. In plants, the median is 1.26 with a range of typical values between 0.87 and 2.12.

Longevity is positively skewed, with rare individuals experiencing an unusually long life. In animals, the median skewness is 2.31 with an interquartile range from 1.99 to 3.08. In plants, the median skewness is 4.31 with typical values ranging from 2.34 to 9.16. These interquartile ranges suggest that distributions of longevity for plants are typically more skewed than for animals.

The median kurtosis in longevity among animals is 7.30 (4.32—11.10). In plants, the median kurtosis is 26.59 (8.01—133.86). The tails of the distribution of longevity thus appear to be heavier for plants than for animals.

## 4.3 Distributions across populations

Fig 1 shows the distribution of values of the various statistics of LRO, among populations of plants and animals (see S1 Appendix for a similar graph for longevity). To make the patterns more clearly visible, for some statistics the data for plants and animals are trimmed by 20% (eliminating the top and bottom 10% of the values). See [28] for a discussion of trimmed statistics.

Among the populations in our sample, plants exhibit a larger range of the statistics of LRO than animals and appear more skewed in the distributions of these statistics. In other words, plants show more uncertainty in their life history trajectories and the reproductive success they experience along the way than animals do. Additionally, plants, more so than animals, differ among populations in the level of this uncertainty.

## 5 Multidimensional life-history strategies

The various demographic outcomes do not vary independently among populations. We compute Pearson product-moment correlations to illustrate this (see S1 Appendix). To do so, variables were log-transformed except for those that could take on zero or negative values, i.e. the kurtosis of LRO in animals, the kurtosis and skewness of longevity in both animals and plants, and the measures of variability of age at maturity for animals and plants. The correlations among the demographic outcomes are shown in S1 Appendix. Means and variances for the same life history trait are often strongly positively correlated. Skewness and kurtosis values are positively correlated with all other skewness and kurtosis values, and are negatively correlated to mean LRO and longevity. That is, longer-lived species, and species with high levels of reproduction tend to exhibit less reproductive uncertainty.

To move beyond pairwise correlations, we note that our analysis characterizes life histories by 16 demographic outcomes; these differ widely among plant and animal species. The life history of each species can be represented by a point in a 16-dimensional trait space. To reveal the patterns of association among traits, we used principal component analysis (PCA) to extract

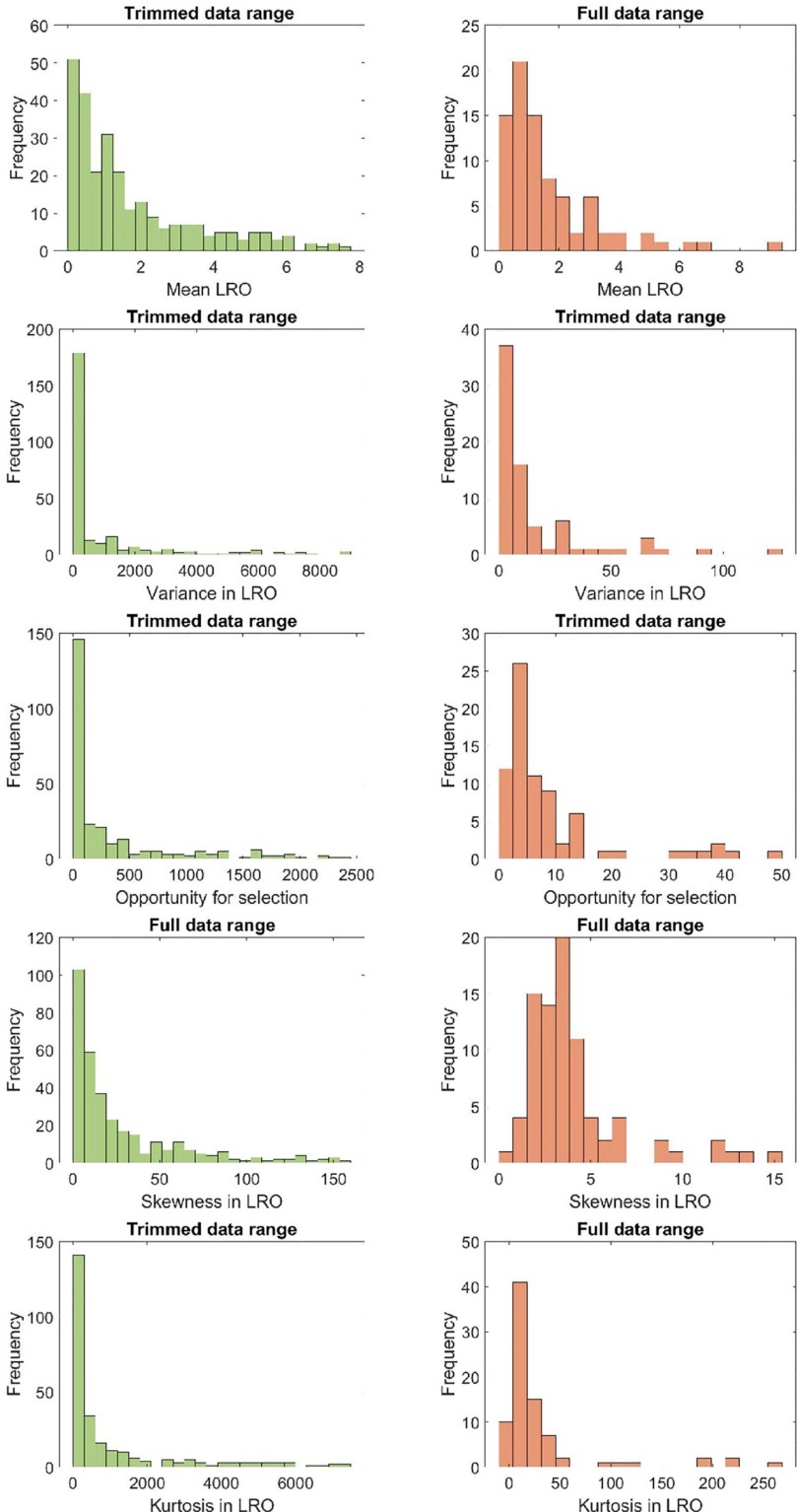

**Fig 1. Histograms of the statistics of LRO (mean, variance, OFS, skewness, and kurtosis) for 332 populations of plants (left hand figures), and for 83 populations of animals (right hand figures).** For both, some distributions were trimmed to better show the shape of the distribution; where the top of the plot reads "Trimmed data range" we left out 20% of the values for animal and plant populations (highest and lowest 10%).

the major axes of variation. Principal component analysis reduces the dimension of the life history space into a set of orthogonal axes that best explain the variation among populations in these outcomes [18–20, 29, 30].

Stearns applied principal component analyses to two sets of data on life history traits in 65 and 162 species of mammal, respectively, and concluded that their life histories are organized along an axis with fast life history strategies on one end and slow ones on the other (i.e. the fast-slow continuum) [18]. Gaillard and collaborators carried out a similar analysis on 80 mammal species and 114 bird species, which concluded that the variation of life history strategies of mammals and birds can be equally described with the fast-slow continuum [29]. They found a secondary gradient corresponding to iteroparity. Salguero-Gómez and collaborators used PCA to analyze the life histories of 418 species of plants. They found two major axes of variation: an axis corresponding to the fast-slow continuum and one reflecting reproductive strategies [19]. Capdevila and collaborators compared 685 terrestrial species and 122 aquatic species of animals and plants, where they also found that the same axes as in [16] explain the variation across populations [30]. None of these analyses included measures of variance, skewness, or kurtosis in the set of life history traits. Healy and collaborators analysed 285 populations, representing 121 animal species, where they also found two axes explaining the differences across population [20]. However, here one axis relates to the fast-slow continuum and the second one to reproduction and the distribution of mortality risk. So far, this is the only study that used a variability measure (i.e. standard deviation of age-specific mortality risk) among the analyzed life-history traits. Our study explicitly focuses on the inclusion of multiple measures of variability in the set of life history traits, as well as the inclusion of a third life-history axis.

We performed PCA analyses on the demographic outcomes after scaling them in such a way that the mean = 0 and SD = 1, as the demographic outcomes are expressed in different units.

## 5.1 Animals

The first three PC axes account for 36.4%, 28.2% and 13.2% of the variance, for a total of 77.8%. The loadings on each of the axes are given in Table 2. Fig 2 shows the distribution of animal species across the three axes, and the loadings of the most important demographic outcomes. The first axis separates species based on statistics related to uncertainty in LRO and longevity, with positive loadings for the opportunity for selection in LRO, and skewness and kurtosis in both LRO and longevity. High scores represent populations with high inter-individual variation relative to the mean of LRO, and a higher chance of extreme values in both LRO and longevity. We refer to this as the life cycle uncertainty axis.

The second axis captures variation in life cycle length. The mean and standard deviation of generation time, as well as standard deviation in longevity, had the highest loadings on the second axis. We refer to this as the life cycle length axis.

The third axis separates populations on the basis of the variability of timing of reproduction in the life cycle, with high loadings for the standard deviation and coefficient of variation of age at maturity, and the coefficient of variation in longevity. Populations at the low end of this axis have low variation in the time to reproduction and time to death relative to the mean. We refer to this as the timing of reproduction axis.

## 5.2 Plants

The first three PCA axes for plants account for 28.2%, 22.9% and 11.9% of the variance, for a total of 63%. The loadings on each of the axes are given in Table 2. Fig 3 shows the distribution

**Table 2. Loadings of lifetime reproductive output (LRO), longevity (LNG), generation time (GT), modes of parity (Parity), and age at maturity (AM) on the first three principal components of animals and plants.**

| Demographic outcome | Animals | | | Plants | | |
|---|---|---|---|---|---|---|
| | PC1 | PC2 | PC3 | PC1 | PC2 | PC3 |
| LRO mean | -0.21 | 0.21 | -0.28 | 0.16 | 0.05 | 0.05 |
| LRO var | 0.08 | 0.15 | -0.36 | 0.10 | -0.15 | -0.14 |
| LRO OFS | **0.37** | 0.12 | -0.21 | -0.17 | **-0.44** | -0.02 |
| LRO skew | **0.39** | 0.14 | -0.08 | -0.17 | **-0.45** | -0.07 |
| LRO kurt | **0.37** | 0.18 | -0.09 | -0.17 | **-0.43** | -0.02 |
| LNG mean | -0.26 | 0.34 | -0.14 | 0.28 | 0.02 | 0.23 |
| LNG sd | -0.24 | **0.37** | -0.09 | **0.37** | -0.007 | 0.24 |
| LNG cv | 0.09 | 0.19 | **0.44** | 0.33 | -0.08 | -0.001 |
| LNG skew | **0.36** | 0.13 | -0.17 | -0.04 | -0.38 | 0.29 |
| LNG kurt | **0.35** | 0.15 | -0.23 | -0.08 | -0.35 | 0.34 |
| GT mean | -0.14 | **0.43** | 0.009 | **0.42** | -0.16 | 0.02 |
| GT sd | -0.16 | **0.41** | -0.04 | **0.41** | -0.13 | 0.16 |
| Parity | -0.16 | 0.11 | 0.11 | 0.12 | 0.03 | **0.51** |
| AM mean | 0.001 | 0.32 | 0.14 | 0.27 | -0.19 | **-0.43** |
| AM sd | 0.19 | 0.20 | **0.43** | 0.30 | -0.18 | **-0.39** |
| AM cv | 0.18 | 0.19 | **0.46** | 0.18 | -0.05 | -0.19 |

of 332 plant populations across three axes of variation, as well as the demographic outcomes with the highest loading on those axes. The first PCA axis captures life cycle length, with the highest loadings for generation time and standard deviation of the ages at production of offspring, and for the standard deviation of longevity.

The second PCA axis arranges species according to uncertainty in lifetime reproduction, with large negative loadings for the standardized variance, skewness, and kurtosis of LRO. At the negative end of this axis, populations have highly variable, skewed, and less predictable lifetime reproductive output.

The third PCA axis in plants is related to the spread of reproduction across the life cycle. Mean and standard deviation of age at maturity had high negative loadings, and mode of parity had a large positive loading on this axis. Populations at the positive end of this axis mature early and are more iteroparous, whereas populations at the negative end of this axis have a shorter range of reproductive years, and mature later.

## 5.3 Comparisons of animals and plants

Animals and plants are distributed across similar life history axes. The first two axes reflect life cycle length and variance therein, and uncertainty in LRO. The differences lie in the order of the axes, and the fact that the uncertainty axis in animals also reflects uncertainty in longevity. The third axes both relate to timing of reproduction, but in animals this also correlates to variation in length of life, whereas in plants the mode of parity plays a larger role. In other words, in plants the third axis reflects the time available to reproduce relative to life cycle length, whereas in animals this axis captures how variable the timing of life cycle events is (Table 3).

Measures of variation in demographic outcomes (variance, scaled variance, skewness, kurtosis) make important contributions to the axes along which life histories of both animals and plants are distributed. It is, arguably, no longer appropriate to ignore individual stochasticity in describing life histories.

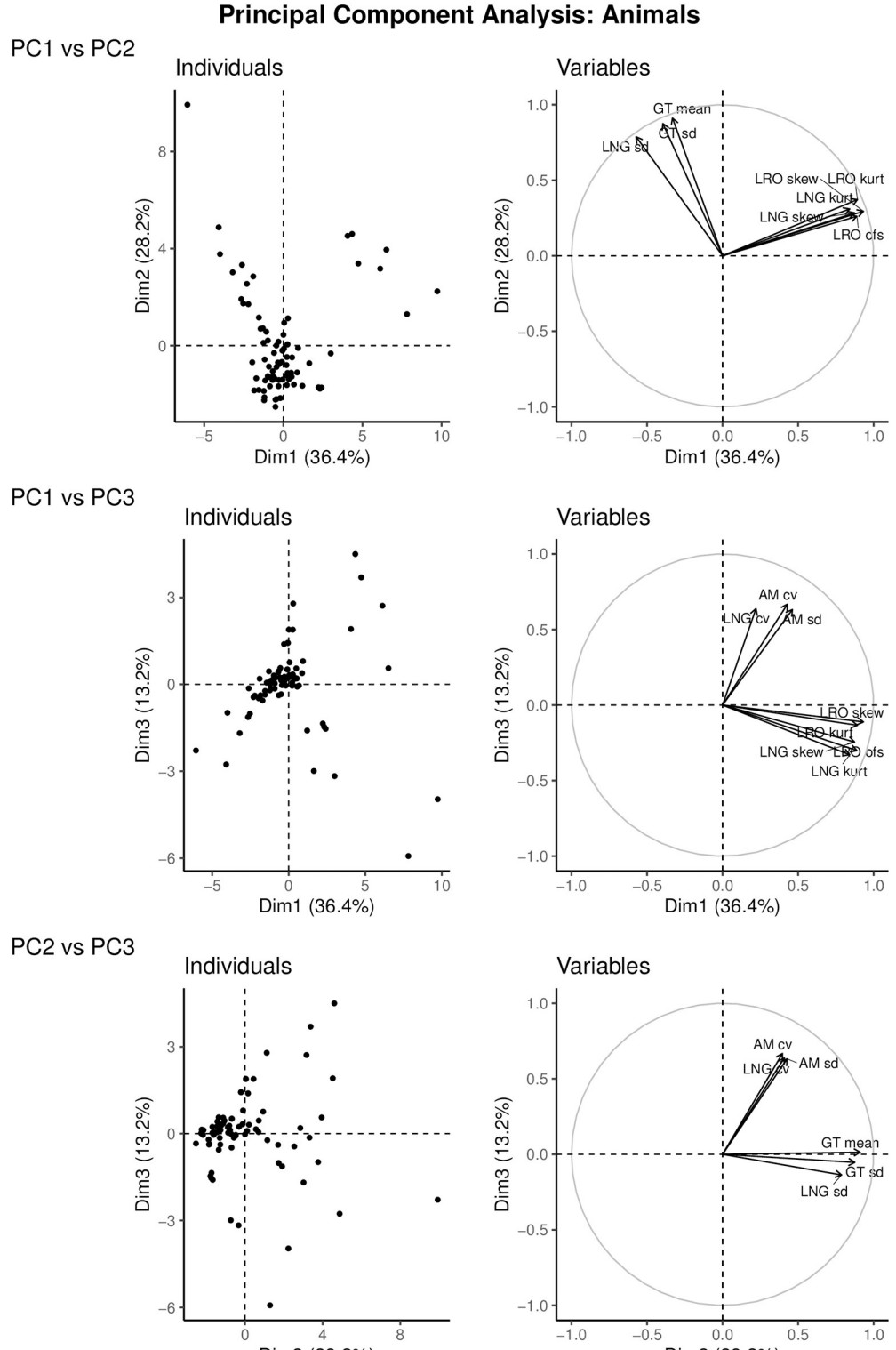

**Fig 2. Principal component axes of 83 populations of animals.** The right-hand graph shows the variables with the highest loadings onto these axes to facilitate visual understanding. Animal populations are primarily distributed along an axis representing measures of uncertainty in LRO and longevity (36.4%), followed by an axis representing generation time and longevity, and variation within these (28.2%). A third axis represents variation in timing of reproduction (13.2%).

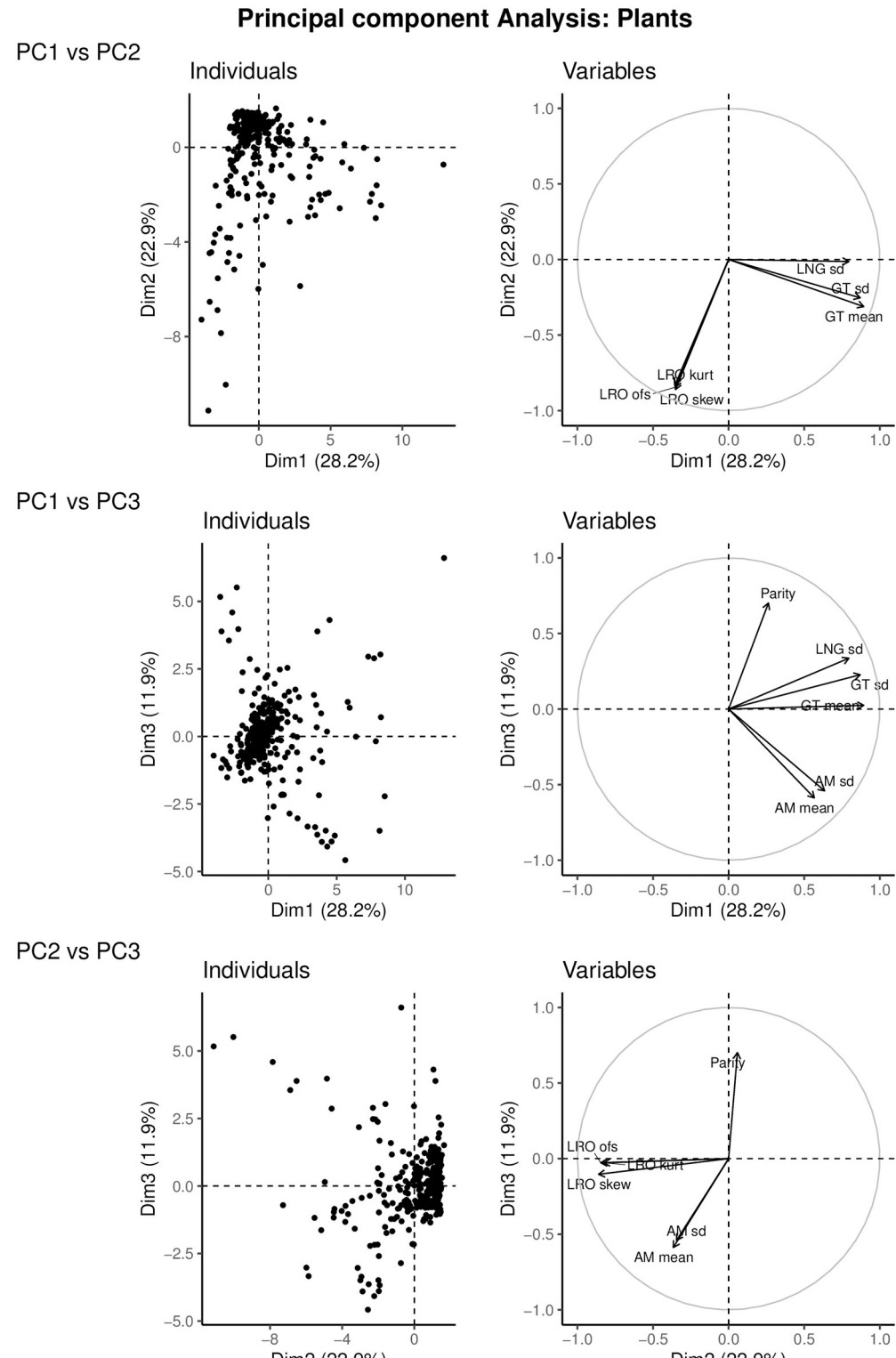

**Fig 3. Principal component axes of 332 populations of plants.** The right-hand graph shows the variables with the highest loadings onto these axes to facilitate visual understanding. Plant populations are primarily distributed along an axis representing generation time and longevity, and variation within these (28.2%), followed by an axis representing measures of uncertainty in LRO (22.9%). A third axis represents mean, variation, and spread of the timing of reproduction (11.9%).

**Table 3. Summary of the three main life-history axes for animals and plants.**

| Axis | Animals | Plants |
|------|---------|--------|
| PC1 | Life cycle uncertainty | Life cycle length |
| PC2 | Life cycle length | Reproductive uncertainty |
| PC3 | Timing of reproduction | Spread reproduction |

# 6 Discussion

Individual stochasticity is a consequence of the life histories of all organisms. It generates variability among individuals in all demographic outcomes. This variability (measured, e.g., by variance, skewness, or kurtosis) is a consequence of the life cycle structure and demographic rates, just as population growth rate, stable stage distribution, the sensitivity and elasticity functions, and other familiar demographic outcomes are consequences. Given Markov chain and MCWR methods, it is now possible to calculate the variability among individuals implied by any life history.

To investigate the resulting patterns requires a sample of a variety of taxa. We have obtained such a sample from the COMADRE and COMPADRE databases. These data are, in a sense, crowd-sourced information on life histories; as such, they provide an overview of the range of published demographic data for plants and animals. Although they are sourced from a scientific crowd (published studies), they are far from being a random or representative sample, either taxonomically or geographically [15, 16]. Even so, they give a picture of the range of what is possible.

The variability created by individual stochasticity is surprisingly large (Table 1, Fig 1). Plants are more variable than animals, in both longevity and LRO. This may reflect the greater flexibility in their life course due to their modular organization. Plant life cycles are often described in ways that provide more alternate pathways through the life cycle than is the case for animals. Shrinkage, for example, is common in plants, but less so in animals [31]. Plants, perhaps more frequently than animals, often experience extremely high mortality at early life history stages (seedlings). This naturally leads to high levels of variance in lifetimes among individuals.

The variability reported here is due to the individual stochasticity that is implied by the rates in the matrices **U** and **F**. Every individual experiences these rates and probabilities. In contrast, empirical measurements of lifetime demographic outcomes, obtained from longitudinal data on individuals followed through their lives, include both stochasticity and heterogeneity. Not every individual experiences the same rates, because they differ in factors not included in the demographic model. Thus the variance in the empirically measured outcomes reflects both stochasticity and whatever heterogeneity may exist among the individuals.

The standardized variance in LRO, which measures the opportunity for selection [23, 32, 33], has been measured empirically and provides a useful comparison to our studies. Robbins and collaborators reviewed empirical estimates of OFS for 22 species of birds and mammals [34]. Their examples have a median of 0.55 and an interquartile range of 0.41–1.17. In 18 human populations, the range is even narrower, with a median of 0.34 and an interquartile range of 0.16–0.46 [35]. This agrees rather well with the standardized variance calculated for a set of 40 developed countries analyzed by [13, 36].

Our results clearly show more variance than these studies report, although our animal sample contains 3 populations within the IQR of [34]. Whether this difference stems from the fact that these empirical studies mostly contain large mammals, or from issues with incorporating nulliparity in empirical studies (e.g., [36, 37]), we cannot say. Potentially, the matrices we

obtained lack certain processes that reduce inter-individual variation, although cases where additional population states reduce variance have not yet been reported. A theoretical example that shows that additional structure in the population result in lower variance would be the variance in longevity in humans. If every individual has the same mortality rate $\mu$, regardless of age (i.e. an alive-dead model), longevity has an exponential distribution of longevity with a standard deviation equal to the mean ($1/\mu$). A typical human population with life expectancy of 70 years would show an SD of 70. But actual human populations, with strongly age-structured mortality, have a SD of longevity on the order of 10 years.

Stochasticity generates variance and uncertainty in life history outcomes such as LRO, longevity, age of reproductive maturity, and generation time. Beyond means, this study presented measures of variation and uncertainty in such life history traits for 332 plant and 83 animal populations. These measures of variability contribute to the life history strategy of a given population. Populations of plants and animals differ, not only in how fast they live, or what reproductive strategy they use, but in how flexible and uncertain their life histories are.

The surprising magnitude of the variation due to individual stochasticity, and its differences among species, cries out for its incorporation into the definition of the space within which life histories are distributed. This multidimensional space is defined by the principal component analyses axes. The first three axes account for 78% of the variance in animal, and 63% of the variance in plant, life histories. This is a step beyond the one-dimensional fast-slow life history continuum which classifies life histories along a continuum from living fast, dying young and producing many offspring to living long and prospering by reproducing slowly but steadily [17, 18]. Later studies have added developmental rate [18], iteroparity [29], and reproductive strategy [19, 20, 30, 38] as secondary axes of variation in life history strategy.

The life history strategies we identify reflect the inherent individual stochasticity in the life cycle. In both animals and plants, the first two axes resemble to some degree the fast-slow continuum and the reproductive strategies axis, but with the crucial distinction that these axes are dominated by inter-individual variation in demographic outcomes. The axis of life cycle length we find (first axis in plants, second axis in animals) incorporates the mean age at offspring production as generation time, and the standard deviations in age at offspring production and longevity. The uncertainty axis (second axis in plants, first in animals) reflects the standardized variance, skewness, and kurtosis in lifetime reproductive output. In animals, this axis furthermore relates to uncertainty in longevity.

Animal and plant populations differ, other than in percentage of variation explained in the first two axes, in a subtle way with regard to the third axis. The third axis, in both animals and plants, relates to the timing of reproduction in the life cycle. For animals, the third axis reflects variability in age at maturity, and standardized variability in longevity. In plants, the third axis reflects the start of the reproductive lifespan in mean and standard deviation of age at maturity, and the length of reproductive lifespan in mode of parity.

From these axes we ascertain that life history trade-offs and constraints are expressed not only in terms of means, but also in degrees of variation among individuals. Inter-individual variation is an important part of the life history strategy, and, therefore, cannot be ignored. This inter-individual variation, however, is completely due to individual stochasticity. Individual heterogeneity is another source of variation in life history outcomes.

If a putative source of heterogeneity is incorporated into the life cycle along with the original individual states, the variance calculated from the resulting model will reflect both sources. The variance in any outcome can then be decomposed into contributions of these sources of variance [39]. Studies that have done so generally find a large, or even overwhelming contribution of stochasticity. Only 5–10% of the variance in longevity in human populations could be attributed to heterogeneity in frailty [5] or socio-economic heterogeneity [40]. In other studies

of longevity, a median of 35% of the variance could be attributed to unobserved heterogeneity in several laboratory studies of insects [41]. About 6% is due to unobserved heterogeneity in the Southern Fulmar [42].

Studies of variance components in lifetime reproductive output are rarer. Snyder and Ellner attributed about 39% of the variance in LRO to heterogeneous 'quality' in Kittiwakes [43]. Jenouvrier and collaborators attributed 22% of the variance in LRO to unobserved heterogeneity in the Southern Fulmar [42]. An analysis of the perennial herb *Lomatium bradshawii* in a stochastic fire environment found that the environment at birth could account for only 0.4% of the variance in LRO. [14]. The broader life history effects of incorporating heterogeneity into models is yet unknown, as these multistate models are still rare.

Our selections from COMADRE and COMPADRE include a wide range of species and life histories, from treecreepers to elephants, from herbs to trees. What if we treat these differences as an extreme case of heterogeneity? How much of the variance in longevity and LRO would be contributed by heterogeneity among populations of different species, in the face of the variance contributed by individual stochasticity within populations?

In general, the variance in some demographic outcome $\xi$ is given by (e.g., [44])

$$V(\xi) = \underbrace{E[V(\xi|\text{population})]}_{\text{within}} + \underbrace{V[E(\xi|\text{population})]}_{\text{between}}.\tag{3}$$

The within-population variance is the contribution of individual stochasticity, since every individual within a population is subject to the vital rates of that population. The between-population variance is due to heterogeneity since it measures the extent to which the population differ in their rates. The contribution of heterogeneity is measured by the intraclass correlation coefficient [45],

$$\mathcal{K} = \frac{V_{\text{between}}}{V_{\text{between}} + V_{\text{within}}}\tag{4}$$

Applying this analysis to our results show that within-population variability due to individual stochasticity in movement through the life cycle overwhelms variation between populations Table 4. This is most apparent in the decomposition of variance in LRO in plants, where less than 1% of the variance is due to population identity, and 99% is due to individual stochasticity within populations. In animals, population identity underlies a greater part of the variance, but still only about 11%. Individual stochasticity determines 89% of the variation. The relative contribution of "heterogeneity" of course depends on both the mean and variance, with the variance between means giving the between-group component, and the mean of variances giving the within-group component. The difference between plants and animals then becomes clear; whereas plants and animals have a very similar median of mean LRO, plants have a 10-fold higher median of variance in LRO than animals (Table 1).

A few extensions of the analysis here that could be applied in a comparative fashion. The analysis of LRO here can be applied to multiple kinds of offspring [8]. This would be

**Table 4. Variance decomposition analysis of animals and plants.** Results show the variance within population ($V_w$), between population ($V_b$), and intraclass correlation coefficient ($\mathcal{K}$) for longevity (LNG) and lifetime reproductive output (LRO).

| Variance | Animals | | Plants | |
|---|---|---|---|---|
| | LNG | LRO | LNG | LRO |
| $V_w$ | 127.36 | 23.62 | $1.03 \times 10^3$ | $3.3 \times 10^3$ |
| $V_b$ | 60.34 | 2.79 | 335.25 | 32.14 |
| $\mathcal{K}$ | 0.3215 | 0.1058 | 0.2453 | 0.0094 |

particularly interesting as a comparison of sexual and clonal reproduction (we have not considered the latter) in plants. The comparisons among species could be informed by sensitivity analysis using results for LRO [4], for longevity and generation time [25], and the intraclass correlation coefficient $\mathcal{K}$ [14]. Here we have focused on the "mean" matrix reported in COMPADRE and COMADRE. But it would be interesting to focus in more detail on the response of variability in demographic outcomes to environmental factors and stressors, as has been done to compare terrestrial versus aquatic species [30] and demographic resilience [46]. Finally, the analysis here treats the matrices **U** and **F** as descriptions of a constant environment, but in cases where data are available, the calculations can be carried out in stochastic environments [8, 14].

In conclusion, large amounts of inter-individual variation is to be expected, because of individual stochasticity within the life cycles of plants and animals. Claims that such variation is due to heterogeneity, especially heritable heterogeneity, require careful support. Measures of variability in longevity, lifetime reproductive output, generation time, age at maturity, and modes of parity should become standard components of life history studies. A diverse collection of species of plants and animals have evolved life histories that reflect both the mean and the variability of life history outcomes.

## Supporting information

**S1 Appendix. Appendix.** Contains further information regarding calculations of demographic outcomes, distributions of summary statistics and correlations among life history outcomes in our sample.
(PDF)

## Acknowledgments

We acknowledge the comments and suggestions from members of the Theoretical Ecology Group of the Institute of Biodiversity and Ecosystem Dynamics, University of Amsterdam.

## Author Contributions

**Conceptualization:** Pablo José Varas Enríquez, Silke Van Daalen, Hal Caswell.

**Data curation:** Pablo José Varas Enríquez.

**Formal analysis:** Pablo José Varas Enríquez, Silke Van Daalen, Hal Caswell.

**Funding acquisition:** Pablo José Varas Enríquez, Silke Van Daalen, Hal Caswell.

**Investigation:** Pablo José Varas Enríquez, Silke Van Daalen, Hal Caswell.

**Methodology:** Pablo José Varas Enríquez, Silke Van Daalen, Hal Caswell.

**Project administration:** Hal Caswell.

**Supervision:** Silke Van Daalen, Hal Caswell.

**Visualization:** Pablo José Varas Enríquez, Silke Van Daalen.

**Writing – original draft:** Pablo José Varas Enríquez, Silke Van Daalen, Hal Caswell.

**Writing – review & editing:** Pablo José Varas Enríquez, Silke Van Daalen, Hal Caswell.

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
