## [Decision Letter · Decision Letter 0]

9 Aug 2022

Individual stochasticity in the life history strategies of animals and plants

PONE-D-22-17310

Dear Dr. Varas Enríquez,

We are pleased to inform you that your manuscript has been judged scientifically suitable for publication and will be formally accepted for publication once it meets all outstanding technical requirements. Congratulations!

The reviewers had a couple small suggestions that you should consider for the final version. 

Kind regards,

Tim A. Mousseau

Academic Editor

PLOS ONE

Reviewers' comments:

Reviewer's Responses to Questions

**Comments to the Author**

1. Is the manuscript technically sound, and do the data support the conclusions?

Reviewer #1: Yes

Reviewer #2: Yes

2. Has the statistical analysis been performed appropriately and rigorously? 

Reviewer #1: Yes

Reviewer #2: Yes

3. Have the authors made all data underlying the findings in their manuscript fully available?

Reviewer #1: Yes

Reviewer #2: Yes

4. Is the manuscript presented in an intelligible fashion and written in standard English?

Reviewer #1: Yes

Reviewer #2: Yes

5. Review Comments to the Author

Reviewer #1: The authors investigate higher ordered moments of life history characteristics for a large number of matrix population models with a focus on differentiating among animal and plant species. The findings of high within population variability in such characteristics, compared to among population variability might be surprising to many field biologists. I have only a few comments that I detail below.

The only comment that I thought would be worth of some additional exploration, is on potential biases generated by structural differences among plant and animal matrixes, and their potential confounding effects with the estimated moments. These structural differences might have less to do with the underlying biology but rather with limitations in data collection. You choose matrixes with >5 stages, among the selected matrixes do they differ in stage numbers (plants vs. animals) and might this create some bias? Higher number of stages greater variances?

Similarly, animals might more often only have sparse matrixes compared to plant matrixes, more matrixes of animals might be age structured only, does such a bias exist and could it influence the findings? Same for reproductive stages; animals might have as a tendency to have less reproductive stages.

A question that might be tricky to address, but relates to the potential issue of sparse matrixes, does the initial number of individuals that have been used to estimate the vital rates differ among plant and animal matrixes? If uncertainty due to small number issues arise it could inflate among population variability (and maybe also within pop variability).

You evaluate age dispersion in reproduction using the Markov chain with rewards method. There are some alternative ways to evaluate and estimate, also for age-stage structured models, such as age dispersion in reproduction (Steiner et al AmNat 2014, Volume 183, Number 6). You could mention these previous directions, not that there is anything wrong using the rewards approach.

Not everyone might know what OFS stands for.

Maybe clarify that survival/mortaltiy and "development" in age ONLY structured populations are not indpendent.

Maybe mention that genotypic differences are expected to generate heterogeneity.

To me it is an open question whether stochastic variability is under selection (a question that goes beyond the manuscript), I agree stochastic variation does not generate heterogeneity selection acts upon in a classical way, but stochastic variation can influence fitness and might be under selection (see e.g. Steiner et al. Scientific reports 11 (1) 1-11.

line 354 typo

The mentioned references are simply meant as examples of various that could be listed.

Reviewer #2: Very sensible analysis -- makes important points about a large collection of life histories. I think these results definitely deserve publication and provide a valuable general perspective on variability, and how important it is.

I have only a few comments; except for the first one they are optional:

1) Page 3, para 2 from bottom: mention that N is defined later!

2) Same place: I suspect these are life histories where the largest stages experience no deaths during the study -- yes or no? May be useful to say which.

3) Page 3, line 2 from bottom -- maybe say more about the "experimentation"!

4) How should the interpreted PCs and the discussion here (and by others) shape our view of life history evolution?

6. PLOS authors have the option to publish the peer review history of their article (what does this mean?). If published, this will include your full peer review and any attached files.

Reviewer #1: **Yes: **Ulrich Steiner

Reviewer #2: **Yes: **Shripad Tuljapurkar

---

## [Editor Report · Acceptance letter]

5 Sep 2022

PONE-D-22-17310 

Individual stochasticity in the life history strategies of animals and plants 

Dear Dr. Varas Enríquez:

I'm pleased to inform you that your manuscript has been deemed suitable for publication in PLOS ONE. Congratulations! Your manuscript is now with our production department. 

Kind regards, 

on behalf of

Dr. Tim A. Mousseau 

Academic Editor

PLOS ONE